# Women’s Access to Sexual and Reproductive Health Services during Confinement Due to the COVID-19 Pandemic in Spain

**DOI:** 10.3390/jcm11144074

**Published:** 2022-07-14

**Authors:** Fatima Leon-Larios, Isabel Silva Reus, Isabel Lahoz Pascual, José Cruz Quílez Conde, María José Puente Martínez, José Gutiérrez Ales, Marta Correa Rancel

**Affiliations:** 1Nursing Department, University of Seville, 41004 Seville, Spain; fatimaleon@us.es; 2Centro de Salud Sexual y Reproductiva de Villena, 03400 Alicante, Spain; isasilreus@gmail.com; 3Hospital Clínico Universitario Zaragoza, 50009 Zaragoza, Spain; 4Hospital Universitario de Basurto, 48013 Bilbao, Spain; jotxe_c@hotmail.com; 5Hospital San Pedro, 26006 Logroño, Spain; mjpuente@riojasalud.es; 6Hospital Universitario Virgen Macarena, 41009 Seville, Spain; jgales050655@gmail.com; 7Hospital Universitario de Canarias, 38320 Santa Cruz de Tenerife, Spain; tenerife1833@gmail.com; 8Departamento de Obstetricia-Ginecología, Pediatría, Preventiva, Medicina Legal y Forense, Microbiología, Parasitología, Universidad de la Laguna, 38200 Santa Cruz de Tenerife, Spain

**Keywords:** contraception, COVID-19, LARC, emergency contraception, reproductive health, sexual health

## Abstract

(1) Background: The COVID-19 pandemic has created a challenge for women’s sexual and reproductive health care. The objective of this research was to analyse access to sexual and reproductive health services during confinement in Spain. (2) Methods: A descriptive and cross-sectional study was conducted with a random sample that was stratified by age in July and August 2020. (3) Results: A total of 1800 women were invited to participate, obtaining a response rate of 98.72%. The frequency of sexual relationships reduced during confinement. Counselling was offered both in person (30.7%) and telematically (39%), although there were also women who experienced access problems (30%). Of those women who required some LARC, only half had access to it, mainly due to the contact difficulties as a result of the limited number of consultations with their prescribing physicians. The use of emergency oral contraception and the use of services for the voluntary interruption of pregnancy were considerably reduced. The women who stated having been victims of gender-based violence were those that lived with their aggressors during confinement and had children. (4) Conclusions: It is necessary to ensure sexual and reproductive rights in confinement times and, for such a purpose, telemedicine can be a good tool that helps to avoid unmet contraceptive needs and unplanned pregnancies.

## 1. Introduction

On 14 March 2020, the state of alarm that imposed confinement of the population in Spain for 3 months was declared, limiting the provision of health services to those considered indispensable. According to a number of published studies, access to sexual and reproductive health services was affected in most countries worldwide [1,2]. The health-crisis situation that was created by COVID-19 produced an unprecedented scenario in health care, also exerting an impact on many women’s sexual and reproductive health [3].

This new situation required scientific associations to adopt a stance on what would be considered unpostponable care provision. Consequently, in Spain, it was considered that the voluntary interruption of pregnancy, emergency contraception, suspected sexually transmitted infections, suspected severe complications associated with different contraceptive methods, and the care provided to victims of gender-based violence would be indispensable services. Undoubtedly, the challenge was related to counselling and the prescription of contraceptive methods via telephone calls or email messages [4,5,6], as well as access to contraceptive methods [3].

The report by the UN Population Fund had already warned that the measures used to fight against COVID-19 might leave 47 million of women without access to contraceptive methods worldwide [7,8]. In fact, the specific outcomes that are related to sexual and reproductive health in Spain reveal some of these challenges. The sale of contraceptive products recorded a strong reduction in the entire country.

The recommendation for the use Long-Acting Reversible Contraception (LARC) was that of choosing an alternative method and postponing the insertion of the IUD/implant. In relation to the withdrawal of these methods, the possibility of postponing extraction was considered due to the chance of extending it a little further based on need, and when their effectiveness could not be ensured, as well as concomitant use of a barrier method [8]. However, this policy was not adopted in all countries, as it was considered to be an essential service in some [7,9,10,11,12].

Predictions indicated that it might be a time in which a baby boom occurred, due to the limitation that some women might face in accessing contraceptive methods; however, according to the Spanish National Statistics Institute [13], no such increase took place, and the birth rate was reduced in the country [14].

On the other hand, the restrictions regarding displacement that were derived from the legislation that was in force at the time compelled women to spend the entire day at home, with many of them subject to potentially hazardous situations that may have been amplified due to the economic hardships faced by those who lost or had to adjust their jobs [3,15,16].

This research aims to analyse the events that took place regarding access to contraception methods and sexual and reproductive health services during the confinement that was caused by the SARS-CoV-19 pandemic in Spain.

## 2. Materials and Methods

A descriptive and cross-sectional study was conducted by means of a hetero-administered telephone survey managed by the SigmaDos survey company (Madrid, Spain). SigmaDos is an international Marketing and Public Survey Study company headquartered in Spain. Participants were contacted on their mobile and landline numbers. The participants were women aged between 15 and 49 years old and living in Spain. Those women who could not communicate in fluent Spanish were not included. Random sampling was performed by age quotas to ensure representativeness of the sample by age and region. The sample size was 1801 participants, with a possible error of +2.35% for a 95.5% confidence level (Sigma Two) and p/q = 50/50.

The data were collected in July and August 2020. The questionnaire included questions that were related to sociodemographic variables, namely: age, schooling level, relationship situation, income level, religious beliefs and nationality. In addition, the participants were asked questions that were related to their sexual habits and access to sexual and reproductive health services during confinement. The questionnaire was prepared by health professionals who specialized in sexual and reproductive health linked to Sociedad Española de Contracepción.

### 2.1. Statistical Analysis

For the categorical variables, the contingency tables were presented according to frequency and percentage values. Contingency tables with Χ^2^ tests were prepared to examine the associations between categorical variables, and ϕ was reported to describe the effect size in each interrelation. A comparison of the means was performed by using the Student’s test. SPSS for Mac (IBM Corp., Armonk, NY, USA), version 23.0, was used for the statistical analysis, considering *p*-values below 0.05 as significant.

### 2.2. Ethical Aspects

The women were free to stop answering questions at any moment. The Spanish health authorities do not require approval by a Research Ethics Committee for this type of study, in which the participants are asked to provide data about their sexual and contraceptive practices outside a health environment that establishes a professional relation. Confidentiality and anonymity were observed in the data treatment, and this is why IRB approval was not required. This study was conducted in accordance with the Declaration of Helsinki.

## 3. Results

A total of 1801 women took part in the study, with a final response rate of 98.72%, which represented 1778 women. The participants’ sociodemographic characteristics are presented in Table 1. These characteristics are similar to the female general population in Spain. Most women lived with their partner, possessed a medium level of education, Spanish nationality and an average-low income level.

### 3.1. Sexual Habits

A total of 45% of the participants stated that the frequency of their sexual relationships was affected by the confinement during the pandemic. A total of 23.6% had not engaged in sexual relationships and 21.4% had reduced their frequency. However, 42% of the women were not affected by the new situation in terms of the frequency of their sexual relationships. The usual frequency in terms of sexual relationships was mostly maintained during confinement among the women who were aged at least 35 years old and in those that lived with their partners. It was precisely in these latter women where the highest increase was recorded (15.2%). A reduction was observed among the women who were aged less than 35 years old [Χ^2^(24) = 136.33, *p* < 0.001, φ = 0.283], and in those that did not live with their steady partners [Χ^2^(8) = 730.31, *p* < 0.001, φ = 0.656], as presented in Table 2.

### 3.2. Access to Contraceptive Methods

A total of 94% of the women had no need to resort to counselling or consultations about contraceptive methods during the confinement period, whereas 5% of the interviewees stated that they needed to access contraceptive counselling, as can be seen in Figure 1; of these, 30% failed to receive it, 39% were not able to access the service via telephone calls, and 30.7% did so in-person.

A total of 93.8% stated not having had any difficulty using or obtaining their usual contraceptive method during confinement. However, 4.4% of the women did experience such inconvenience. The most frequent reason why they faced difficulties to access their usual contraceptive method during confinement was not being able to renew the prescription due to the impossibility of contacting their prescribing physicians (63.3%), followed by dispensation problems in the pharmacies (18.3%). In addition, it was observed that those women who had more problems obtaining their usual method belonged to the age groups from 25 to 29 years old and from 30 to 34 years old: Χ^2^ (12) = 30.51, *p* = 0.002, φ = 0.134.

A total of 95.1% of the women did not need to access any long-acting contraceptive method (IUD, implant) during confinement. Approximately half (48.15%) of the 3% that required a LARC method during this period were not able to access it.

A total of 2.8% of the women had to access and use the morning-after pill. This demand exceeded 5% among the women belonging to the age groups from 25 to 29 years old and from 35 to 39 years old: Χ^2^(12) = 25.79, *p* = 0.011, φ = 0.123, as can be seen in Table 3. However, this is a considerably lower percentage than the habitual use rate in Spanish women, which is generally around 30–40%.

### 3.3. Voluntary Interruption of Pregnancy

During 2020, the voluntary interruption of pregnancy (VIP) also recorded a decrease (17%), which represented a marked reduction during the confinement months. It was necessary to conduct the counselling sessions telematically and, frequently, to attend the clinic only once. Only Catalonia and Galicia implemented this by discontinuing the in-person counselling visits during confinement. It is not possible to establish if this decrease is related to access to services of VIP, or to the fact that less women needed the service.

### 3.4. Gender-Based Violence

During confinement, 2.2% of the women stated having experienced some situation of gender-based violence, especially those with lower schooling levels (7.2%) [Χ^2^(4) = 12.44, *p* = 0.014, φ = 0.086]; lower income levels (4.7%) [Χ^2^(3) = 16.59, *p* = 0.001, φ = 0.105]; with children (12.4%) [Χ^2^(6) = 12.63, *p* = 0.049, φ = 0.086]; and those who were practicing Catholics (8%) [Χ^2^(6) = 44.79, *p* < 0.001, φ = 0.162], as presented in Table 4. However, no differences were observed among the women who lived with their partners [Χ^2^(4) = 4.32, *p* = 0.364]; by different age groups [Χ^2^(12) = 17.54, *p* = 0.130]; or by nationality [Χ^2^(2) = 0.27, *p* = 0.874], being not statistically significant.

## 4. Discussion

This research aimed at knowing women’s experiences in Spain during the COVID-19 pandemic, regarding access to sexual and reproductive health services during the decreed confinement period. Questions were asked about access and counselling regarding usual contraception and emergency contraception, as well as whether they had experienced situations of gender-based violence.

Confinement ushered in some changes in women’s lifestyles, with their sexual habits among them. As is the case in other studies, our results indicated that the frequency of sexual relationships was reduced among women [17,18], being even lower in those that did not live with their partners.

In our study, 5% of the participating women required contraceptive counselling during the pandemic, and of them, 3 out of 10 were not able to access it due to difficulties contacting their usual physicians, as already pointed out in the studies by Manze et al. [2] in the USA and by Dema et al. [18] in the UK. These data are relatively higher than the data that were identified in Sweden, where few women had difficulties accessing contraceptive prescriptions and/or counselling [19]. However, these results must be interpreted considering that full confinement of the population was not implemented in Sweden, as was the case in Spain. Nevertheless, a reduction in the prescription of LARC methods was in fact identified in the aforementioned study, a result that is in line with what happened in Spain, with the possibility of having occurred in a higher number of unplanned pregnancies [19,20,21]. This might be because, unlike the case in other countries, it was not defined as an essential service in Spain [7].

Age was identified among the factors that were described in other studies, as well as people who reported risk behaviours as a risk factor for facing difficulties accessing the services [18,22].

In our research, the women who reported more difficulties accessing contraceptive methods were those belonging to the age group from 25 to 34 years old, coinciding with users of hormonal or LARC contraceptive methods. Our results indicated that 6 out of 10 women were able to access counselling on contraception with a health professional, whether in-person or via telephone calls. Use of the new technologies that comprise Telehealth is an alternative that would have to be considered as an option to meet the sexual and reproductive health needs of this collective, and such use should be increased in situations like those that were experienced during confinement, as already pointed out in previous studies [23,24,25]. Studies that have appraised the use of these alternatives to in-person visits have attested to their usefulness to ensure access to the services [18,21]. However, the results arising from this research indicate that some women were not able to have their demand met, as was the case in many countries [21]. However, the digital gap which might be experienced by the most vulnerable collective groups that were not educated on new technologies or lacked good quality access to the Internet should indeed be considered [26].

The use of emergency contraception during the pandemic was considerably reduced to unusual consumption values in Spain, where 3 out of 10 women resorted to emergency oral contraception [27]. Dispensation of emergency oral contraception products is free in pharmacies; therefore, their use and access by women should not have been compromised in Spain. However, these data are in line with what happened in other European countries where consumption was also reduced [28].

In a study that was conducted in 29 countries, access to abortion was limited in most of them during the COVID-19 pandemic [19].

The voluntary interruption of pregnancy was considered as an essential service during the pandemic, and bureaucracy was reduced to ease its provision to the women who required it in many countries [29]. However, the statistics showed a reduction in the number of abortions in Spain, a situation that was similar to that in other European countries [1,30], although an increase in the number of unsafe abortions was also verified [21]. Some European countries set out the proposal to increase pharmacological abortions to facilitate such provision to the women who needed them [31].

As predicted in some studies and reports, expressions of gender-based violence during the pandemic and confinement increased, despite the national statistics from many countries not reflecting such a rise [3,15,16]. The number of victims of gender-based violence was reduced by up to 8.4% during 2020 in Spain; the rate was 1.4 for every 1000 women aged at least 14 years old [32]. The highest reduction was recorded during the first two months of confinement, a trend that was maintained in the second part of 2020 in Spain [32]; however, it is unclear whether this decrease was real or was in fact driven by under-reporting due to the harsher confinement situation that was experienced by the victims with their aggressors. In the sample that comprises our research, 2 out of 100 women stated having experienced some situation of gender-based violence during confinement, with this being more frequent in those who lived with their partners and had children. At the global level, the reality indicated that the number of cases of gender-based violence increased with the COVID-19 pandemic [1].

Some of the strengths of this study lie in the sample size and its representativeness, as participation was high. This research allows identification of the weaknesses that were detected, in order to face them should confinement be implemented again. Among the limitations, we identified that it was not possible to establish a baseline that allowed us to compare the pre-pandemic situation with the experiences during confinement and, thus, establish comparisons. On the other hand, we assessed the access to sexual and reproductive health services, asking women about their perception of the situation experienced during the pandemic, though not based on access statistics.

## 5. Conclusions

The pandemic exerted a negative impact on women’s sexual and reproductive health, exposing them to risk situations for unplanned pregnancies by not having access to the usual services that were provided prior to the pandemic, and that were now limited. Access to LARCs was considerably reduced and delayed. It is necessary to integrate the difficulties that were identified by the women in accessing sexual and reproductive health services, as well as to ensure sexual and reproductive rights through the provision of these services. For such a purpose, the suggestion is to enhance and strengthen Telemedicine with resources that enable an efficient response in situations of confinement or restrictions, regarding access to health services in similar situations in the future.

## Figures and Tables

**Figure 1 jcm-11-04074-f001:**
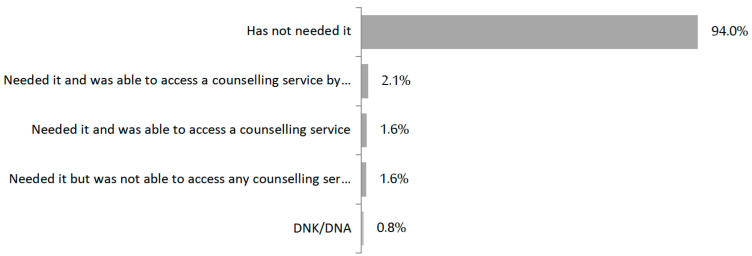
Access to counselling or consultations about contraceptive methods.

**Table 1 jcm-11-04074-t001:** Sociodemographic characteristics.

Variables	Total *n* (%) 1778 (100)
Age	
15–19 years old	170 (9.6)
20–24 years old	189 (10.6)
25–29 years old	211 (11.9)
30–34 years old	249 (14)
35–39 years old	308 (17.3)
40–44 years old	335 (18.8)
45–49 years old	316 (17.8)
Relationship situation	
Lives with her partner	1011 (56.9)
Steady partner, but not living together	325 (18.3)
No steady partner	442 (24.9)
Schooling level	
Elementary school	84 (4.7)
High school	917 (51.9)
University studies	767 (43.4)
Nationality	
Spanish	1602 (90.2)
Other	174 (9.8)
Income level	
Low	401 (25.8)
Average-low	663 (42.6)
Average	300 (19.3)
High	191 (12.3)

**Table 2 jcm-11-04074-t002:** Frequency of sexual relationships during confinement in relation to age and relationship situation.

Sexual Relationships	Total *n* (%)	Age							Relationship Situation		
		15–19 *n* (%)	20–24 *n* (%)	25–29 *n* (%)	30–34 *n* (%)	35–39 *n* (%)	40–44 *n* (%)	45–49 *n* (%)	Lives with Her Partner *n* (%)	With a Partner, but Not Living Together *n* (%)	No Steady Partner *n* (%)
Increased	193 (11.3)	10 (8.5)	19 (10.4)	28 (13.4)	31 (12.6)	44 (14.5)	35 (10.6)	35 (8.3)	152 (15.2)	30 (9.6)	11 (2.8)
Reduced	362 (21.4)	31 (25.6)	56 (30.7)	49 (23.5)	57 (23.3)	51 (17.1)	65 (19.8)	65 (17.2)	175 (17.6)	102 (32.6)	85 (21.9)
Unchanged	169 (42.8)	15 (12.3)	51 (27.9)	88 (42.2)	104 (42.6)	143 (47.8)	157 (47.7)	157 (53.6)	620 (62.0)	63 (20.2)	44 (11.3)
No sexual relationships	400 (23.6)	64 (53.6)	55 (30.1)	44 (21.1)	52 (21.3)	60 (19.9)	66 (20.1)	66 (18.9)	38 (3.8)	117 (37.6)	244 (63.2)
DNK/DNA	17 (1.0)	0 (0.0)	2 (0.9)	0 (0.0)	1 (0.2)	2 (0.6)	6 (1.8)	6 (2.0)	13 (1.3)	0 (0.0)	3 (0.8)

**Table 3 jcm-11-04074-t003:** Use of the emergency contraception pill (ECP) during confinement.

Have You Needed ECP?	Total *n* (%)	Age						
		15–19 *n* (%)	20–24 *n* (%)	25–29 *n* (%)	30–34 *n* (%)	35–39 *n* (%)	40–44 *n* (%)	45–49 *n* (%)
Yes	47 (2.8)	1 (1.1)	5 (3.0)	11 (5.5)	1 (0.3)	17 (5.6)	6 (1.9)	6 (1.8)
No	1646 (96.9)	118 (98.9)	177 (97.0)	197 (94.5)	243 (99.5)	282 (94.0)	323 (97.8)	306 (97.5)
DNK/DNA	0 (0.3)	0 (0.0)	0 (0.0)	0 (0.0)	1 (0.2)	1 (0.4)	1 (0.3)	5 (0.7)

**Table 4 jcm-11-04074-t004:** Factors associated with having being victims of gender-based violence during confinement.

Variables	Have You Been a Victim of Gender-Based Violence?	
	Yes *n* (%) 38 (2.2)	No *n* (%) 1649 (97.8)
Schooling level		
Elementary school	6 (7.6)	73 (92.4)
High school	19 (2.2)	836 (97.8)
University studies	13 (1.9)	740 (98.1)
Income level		
Low	18 (4.7)	366 (95.3)
Average-low	17 (2.5)	635 (97.5)
Average	2 (0.7)	290 (99.3)
High	0 (0)	185 (100)
Religious beliefs		
Practicing Catholic	19 (8)	214 (92)
Lapsed Catholic	8 (1.1)	708 (98.9)
Other religions	2 (1.6)	100 (98.4)
Agnostic/Atheist	9 (1.4)	635 (98.6)
Children		
Yes	21 (2.44)	839 (97.5)
No	17 (2)	817 (98)

## Data Availability

The datasets that support the findings of this study are available from the corresponding author upon reasonable written request.

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
