# Peer review of "Women’s Access to Sexual and Reproductive Health Services during Confinement Due to the COVID-19 Pandemic in Spain"

_jcm, 2022, doi:10.3390/jcm11144074_

Round 1
Reviewer 1 Report
This is an important topic with significant ramifications for providing healthcare and services to women during the pandemic. Some minor editorial/language corrections in the first paragraph, some of which may be English language issues, can be corrected with copy-editing. I note a few of them here, and other more substantive comments for improvement.
Line 29 (abstract): “having being” should be "having been”
“Was declared”- was this a WHO announcement? A specific date is mentioned (march 14th)- not clear if this was specific to region, the world, nation, etc. and no reference.
“In fact, the statistics related to sexual and reproductive health in Spain were affected by…” It could say demonstrate some of these challenges (remove “were affected”) The statistics were not affected- the specific outcomes were affected.
Following are specific questions/suggestions for the text: "On the other hand, the restrictions regarding displacement derived from the legislation in force at the time led many women to spend the entire day at their homes, with the possibility of disruptive situations ... (do you mean relationship challenges?) among couples, that may have been (instead of: could sometimes be) amplified due to the economic hardships faced by many people (who lost or had to adjust their jobs?) .
Line 76- what is “hetero-applied” Does this mean heterosexual couples? Actually not sure at all what it means.
Regarding IRB/research approval: Although I assume that the institution was aware of this, I’m confused about the lack of need for IRB approval. Even if it is “exempt”, would the researchers not have been required to apply for such approval and get this classification? The questions may not have been individually identified, but the topic and nature of the questions might have raised anxiety and thus should have been reviewed by an IRB. This is a major concern. Please clarify the review and consent process.
Was the 98% “final response rate” from among all of the women called by the random sampling, or from among women who agreed to participate and then completed the questions? These are two very different things and makes it appear that 98% of the people who were dialed responded to the survey. This is an unusually high rate for a random phone survey on a sensitive topic.
I would like to see the results section begin with a brief narrative description, “presented in table 1, the study sample was predominantly Spanish, mean age of X, more than half living with a partner, etc “ rather than just: the study sample is described in the table. The reader should have some sense of the overall representativeness, or characteristics of the sample, and how they relate to what was sought.
In the results, it would be useful to see the chi-square table for those features that were significant and important (e.g. those who had difficulty getting their contraceptives by age group, for example seems important and those data for significance should be presented.)
The decrease in “voluntary interruption of pregnancy” was assessed only as a categorical yes or no, and was not explored by access- that is, did fewer women want or need VIP/ or were those who wanted it less able to access it? These are two important but different questions, and if the study does not have the answer to that, then it should be stated as a limitation or as a possibility, rather than as an outcome on its face.
The gender-based violence percentages and significance values mix p<.001s with p values between .05 and .1. There is no real presentation of what is considered important or meaningful. So the items that are p=.086 seem out of place here. There should be some discussion in the methods perhaps of what was considered significant, and then the authors can note that several items had near-significant differences. Or be clear that the cutoff is p=.1
The following line was very confusing: Use of emergency contraception during the pandemic was considerably reduced to unusual consumption values in Spain, where 3 out of 10 women have resorted to emergency oral contraception [27]. Does "Reduced to unusual consumption" mean lesser, because 3 out of 10 women seems high to this ready? But perhaps this meant that 3 out of 10 women who reported needing them was able to get them (i.e. 7 out of 10 who needed them had no access). That’s a very different statement and isn’t clear from this sentence.
As well, I was confused here: As predicted in some studies and reports, the expressions of gender-based violence during the pandemic and confinement were increased despite the fact that the national statistics from many countries did not reflect such rise [3,15,16]. The number of victims of gender-based violence was reduced up to 8.4% during 2020 in Spain; the rate was 1.4 for 231 every 1,000 women aged at least 14 years old [32]. The highest reduction was recorded 232 during the first two months of confinement, a trend that was maintained in the second 233 semester of 2020 in Spain [33]; It appears to say that the rate is increased, but then you say reduced. Did you mean that it was increased in other literature, but in THIS study it was reduced? That is unclear here.
At the end of the discussion, you say the study’s “Strengths lie in representativeness”. This is only the case if, in fact, 98% of the women sampled participated- see my question above about who actually did participate. If it was actually 98% of the women called randomly on the phone, then please describe the methods used to encourage participation, as this is an astoundingly high response rate.
For the final conclusion, I would be more inclined to say that, for the large majority of women, access was not a problem, but for those who it was, the results were severe and problematic. I think the assumption that these results of lack of access are severe is based on assuming that they were pandemic-related and were worse than they were before the pandemic. However, the main limitation is not having data from pre-pandemic, and this limitation is not given the importance that it has. Thus given the relatively low percentage of women who reported poor access, I think the conclusion should be less dire. In fact, it might suggest further exploration of how this compares to non-pandemic times and/or to explore perhaps some open-ended exploration with the women who did not have access, as to whether or not this differed from their pre-pandemic experiences. There is a causal assumption here that cannot be concluded from this cross-sectional data and that is a limitation that is not as clearly presented as it should be, both in the conclusion and in the discussion.
The study overall has great value and interest, but its conclusions should be tempered by its limitations, which are greater than are presented in this manuscript. I think that's really important, as are what the study DID find.
Author Response
This is an important topic with significant ramifications for providing healthcare and services to women during the pandemic. Some minor editorial/language corrections in the first paragraph, some of which may be English language issues, can be corrected with copy-editing. I note a few of them here, and other more substantive comments for improvement.
Response: Thank you for your observations and recommendations.
Line 29 (abstract): “having being” should be "having been”
Response: This suggestion has been considered.
“Was declared”- was this a WHO announcement? A specific date is mentioned (march 14th)- not clear if this was specific to region, the world, nation, etc. and no reference.
Response: This data was clarified. The reference was about Spain.
“In fact, the statistics related to sexual and reproductive health in Spain were affected by…” It could say demonstrate some of these challenges (remove “were affected”) The statistics were not affected- the specific outcomes were affected.
Response: This was amended. Please, see line 54.
Following are specific questions/suggestions for the text: "On the other hand, the restrictions regarding displacement derived from the legislation in force at the time led many women to spend the entire day at their homes, with the possibility of disruptive situations ... (do you mean relationship challenges?) among couples, that may have been (instead of: could sometimes be) amplified due to the economic hardships faced by many people (who lost or had to adjust their jobs?) .
Response: This suggestion has beeb included. Please, see line 70-71
Line 76- what is “hetero-applied” Does this mean heterosexual couples? Actually, not sure at all what it means.
Response: This term was changed to “hetero-administered”.
Regarding IRB/research approval: Although I assume that the institution was aware of this, I’m confused about the lack of need for IRB approval. Even if it is “exempt”, would the researchers not have been required to apply for such approval and get this classification? The questions may not have been individually identified, but the topic and nature of the questions might have raised anxiety and thus should have been reviewed by an IRB. This is a major concern. Please clarify the review and consent process.
Response: Informed consent was obtained from all subjects involved in the study. Written informed consent was obtained from the participants to publish this paper. The study was conducted in accordance with the Declaration of Helsinki. Ethical review and approval were waived for this study due to the fact that the subjects were invited to participate voluntarily in it.
Was the 98% “final response rate” from among all of the women called by the random sampling, or from among women who agreed to participate and then completed the questions? These are two very different things and makes it appear that 98% of the people who were dialed responded to the survey. This is an unusually high rate for a random phone survey on a sensitive topic.
Response: 98% was the final response rate of women who agreed to participated and then completed the questions. When a woman refused to participate, she was replaced by another woman with similar characteristics.
I would like to see the results section begin with a brief narrative description, “presented in table 1, the study sample was predominantly Spanish, mean age of X, more than half living with a partner, etc “ rather than just: the study sample is described in the table. The reader should have some sense of the overall representativeness, or characteristics of the sample, and how they relate to what was sought.
Response: A paragraph with the information was included. Please, see lines 104-106.
In the results, it would be useful to see the chi-square table for those features that were significant and important (e.g. those who had difficulty getting their contraceptives by age group, for example seems important and those data for significance should be presented.)
Response: This information was included in the text before the table to avoid extend the table. Please, see lines 116-119, as an example.
The decrease in “voluntary interruption of pregnancy” was assessed only as a categorical yes or no, and was not explored by access- that is, did fewer women want or need VIP/ or were those who wanted it less able to access it? These are two important but different questions, and if the study does not have the answer to that, then it should be stated as a limitation or as a possibility, rather than as an outcome on its face.
Response: This information was not provided in the study so we could not establish a relation. Though this approach suggested is interesting, it is not possible to incuded it. A clarification about this question has been added.
The gender-based violence percentages and significance values mix p<.001s with p values between .05 and .1. There is no real presentation of what is considered important or meaningful. So the items that are p=.086 seem out of place here. There should be some discussion in the methods perhaps of what was considered significant, and then the authors can note that several items had near-significant differences. Or be clear that the cutoff is p=.1
Response: This aspect was included in the results section. Besides, in the discussion section, included what was relevant (please see line 239).
The following line was very confusing: Use of emergency contraception during the pandemic was considerably reduced to unusual consumption values in Spain, where 3 out of 10 women have resorted to emergency oral contraception [27]. Does "Reduced to unusual consumption" mean lesser, because 3 out of 10 women seems high to this ready? But perhaps this meant that 3 out of 10 women who reported needing them was able to get them (i.e. 7 out of 10 who needed them had no access). That’s a very different statement and isn’t clear from this sentence.
Response: Thank you for this observation. Almost 4 out of 10 women usually access to emergency oral contraception (we have included the reference). However, during the COVID-19 pandemic, this data decreased to 2.8%.
As well, I was confused here: As predicted in some studies and reports, the expressions of gender-based violence during the pandemic and confinement were increased despite the fact that the national statistics from many countries did not reflect such rise [3,15,16]. The number of victims of gender-based violence was reduced up to 8.4% during 2020 in Spain; the rate was 1.4 for 231 every 1,000 women aged at least 14 years old [32]. The highest reduction was recorded 232 during the first two months of confinement, a trend that was maintained in the second 233 semester of 2020 in Spain [33]; It appears to say that the rate is increased, but then you say reduced. Did you mean that it was increased in other literature, but in THIS study it was reduced? That is unclear here.
Response: We have clarified this issue. Gender-based violence during the pandemic and confinement increased, but this increase was not included in statistics because many women under-reported the situation. The reason for that is that these women were afraid of reportin their partners as they spend most of the time confined with their aggressors. We state this issue: “The highest reduction was recorded during the first two months of confinement, a trend that was maintained during the second semester of 2020 in Spain”
At the end of the discussion, you say the study’s “Strengths lie in representativeness”. This is only the case if, in fact, 98% of the women sampled participated- see my question above about who actually did participate. If it was actually 98% of the women called randomly on the phone, then please describe the methods used to encourage participation, as this is an astoundingly high response rate.
Response: This issue was amended previously.
For the final conclusion, I would be more inclined to say that, for the large majority of women, access was not a problem, but for those who it was, the results were severe and problematic. I think the assumption that these results of lack of access are severe is based on assuming that they were pandemic-related and were worse than they were before the pandemic. However, the main limitation is not having data from pre-pandemic, and this limitation is not given the importance that it has. Thus given the relatively low percentage of women who reported poor access, I think the conclusion should be less dire. In fact, it might suggest further exploration of how this compares to non-pandemic times and/or to explore perhaps some open-ended exploration with the women who did not have access, as to whether or not this differed from their pre-pandemic experiences. There is a causal assumption here that cannot be concluded from this cross-sectional data and that is a limitation that is not as clearly presented as it should be, both in the conclusion and in the discussion.
Response: We based this assumption on women´s opinions. Some services related to sexual and reproductive health were limited and suspended. So, we concluded this statement. On the other hand, we concluded that we could not compare pre pandemic and post pandemic in limitations.
The study overall has great value and interest, but its conclusions should be tempered by its limitations, which are greater than are presented in this manuscript. I think that's really important, as are what the study DID find.
Reviewer 2 Report
The authors aim to analyze the access to sexual and reproductive health services during confinement in Spain. The title should mention the study population (women).
This is a promising article. I have some suggestions for your manuscript.
Introduction: The rationale and hypothesis for the study could be less descriptive and more appealing. A more critical discussion of the links between sexual health (a conceptual definition is missing) and assess to sexual health services and related barriers could be presented in the introduction, which should provide sufficient background and include the most relevant references. Your introduction would benefit from presenting the context of sexual health besides Covid-19 in the specific case of Spain and using proper references citations.
A more comprehensive overview of the approaches to sexual and reproductive health diversity of needs across life course and populations would also be expected. It should be inclusive of the needs of women and girls, of different ages, those belonging to national or ethnic groups, and religious and linguistic minorities from different parts of the world with different contexts and experiences. An explicit statement of the objectives being addressed with reference to the concepts, contexts, and population seems to be missing.
I would recommend the following articles:
Ferreira-Filho ES, de Melo NR, Sorpreso ICE, Bahamondes L, Simões RDS, Soares-Júnior JM, Baracat EC. Contraception and reproductive planning during the COVID-19 pandemic. Expert Rev Clin Pharmacol. 2020 Jun;13(6):615-622.
Larki M, Sharifi F, Manouchehri E, Latifnejad Roudsari R. Responding to the Essential Sexual and Reproductive Health Needs for Women During the COVID-19 Pandemic: A Literature Review. Malays J Med Sci. 2021 Dec;28(6):8-19.
Mukherjee TI, Khan AG, Dasgupta A, Samari G. Reproductive justice in the time of COVID-19: a systematic review of the indirect impacts of COVID-19 on sexual and reproductive health. Reprod Health. 2021 Dec 20;18(1):252.
The aim of the study needs clarification. In the abstract v. main text. And should be linked to the results.
Methods need to be described in more detail and clarified. Sampling procedures need more detail. Was there a sampling of both mobile and landline numbers? Also, characterize the survey company and the data collection quality procedures.
How was the study presented to the participants; was there an oral informed consent collected and was there a different procedure for participation among younger than 18 years of age participants? Also, provide participation rate and any comparisons between participants and refusals if possible.
Please indicate eligibility and exclusion criteria. Was language an exclusion criterion?
Measure: Please describe the main outcome measures. Which services were considered, how was the access measured, and was the quality of the services evaluated? Was there a comparison question before and during the confinement period to evaluate the effect of covid-19?
Ethics: Personal data was used (such as the telephone), besides sensitive data (such as nationality, religion, sex life). The project should have been submitted to the Ethics Committee. Please correct the information and provide justification for not having a formal approval. Also, please detail how was confidentiality anonymity preserved. Was there any protocol for sensitive issues?
Please discuss ethical issues beyond confidentiality and data protection. Could you better explain the procedures for the protection of the well-being of the research participants given the specificity of the interview guide and sensitive questions such as gender-based violence and voluntary interruption of pregnancy?
Results: The results and the discussion sections need to be organized to answer the previously defined objectives. Tables and figures need to be improved.
Participants’ characterization could be improved and present more information (region seems to be missing); also consider presenting smaller age groups and social factors comparisons in terms of main outcomes (such as Table 4 – nevertheless age is missing).
The discussion is too descriptive. Also, the authors need to better discuss the study's strengths and limitations.
Conclusion: Implications of the results for public policy should be enlightened. Specific future recommendations for research and action should be added.
In short, given the data collected, I think there is the potential to do more with the analysis.
Author Response
The authors aim to analyze the access to sexual and reproductive health services during confinement in Spain. The title should mention the study population (women).
This is a promising article. I have some suggestions for your manuscript.
Response: Thank you for this observation.
Introduction: The rationale and hypothesis for the study could be less descriptive and more appealing. A more critical discussion of the links between sexual health (a conceptual definition is missing) and assess to sexual health services and related barriers could be presented in the introduction, which should provide sufficient background and include the most relevant references. Your introduction would benefit from presenting the context of sexual health besides Covid-19 in the specific case of Spain and using proper references citations.
Respuesta: Thank you for this suggestion. This is cross-sectional research aims to know more about these circumstances during the pandemic. Our intention was not to stablish an hypothesis in advance. We focused all this paper on what happened in this specific time in Spain.
A more comprehensive overview of the approaches to sexual and reproductive health diversity of needs across life course and populations would also be expected. It should be inclusive of the needs of women and girls, of different ages, those belonging to national or ethnic groups, and religious and linguistic minorities from different parts of the world with different contexts and experiences. An explicit statement of the objectives being addressed with reference to the concepts, contexts, and population seems to be missing.
Response: We did not find that aspects such as age, ethnicity or religion had an influence on this particular experience. For this reason, the above factors were not discussed.
I would recommend the following articles:
Ferreira-Filho ES, de Melo NR, Sorpreso ICE, Bahamondes L, Simões RDS, Soares-Júnior JM, Baracat EC. Contraception and reproductive planning during the COVID-19 pandemic. Expert Rev Clin Pharmacol. 2020 Jun;13(6):615-622.
Larki M, Sharifi F, Manouchehri E, Latifnejad Roudsari R. Responding to the Essential Sexual and Reproductive Health Needs for Women During the COVID-19 Pandemic: A Literature Review. Malays J Med Sci. 2021 Dec;28(6):8-19.
Mukherjee TI, Khan AG, Dasgupta A, Samari G. Reproductive justice in the time of COVID-19: a systematic review of the indirect impacts of COVID-19 on sexual and reproductive health. Reprod Health. 2021 Dec 20;18(1):252.
The aim of the study needs clarification. In the abstract v. main text. And should be linked to the results.
Response: This was reworded. Please see lines 73-75
Methods need to be described in more detail and clarified. Sampling procedures need more detail. Was there a sampling of both mobile and landline numbers? Also, characterize the survey company and the data collection quality procedures.
Response: Thank you for your observation, this issue was clarified. Please see line 78-80
How was the study presented to the participants; was there an oral informed consent collected and was there a different procedure for participation among younger than 18 years of age participants? Also, provide participation rate and any comparisons between participants and refusals if possible.
Response: Thank you for this observation. All participants were informed about the aims of the research. Informed consent collected was requested and collected prior to participation (lines 269-270).
Please indicate eligibility and exclusion criteria. Was language an exclusion criterion?
Response: We appreciate this observation, that has been included: Please see lines 80-82.
Measure: Please describe the main outcome measures. Which services were considered, how was the access measured, and was the quality of the services evaluated? Was there a comparison question before and during the confinement period to evaluate the effect of covid-19?
Response: This observation is really interesting, but sadly, this could not be assessed and thus we cannot provide information in this sense.
Ethics: Personal data was used (such as the telephone), besides sensitive data (such as nationality, religion, sex life). The project should have been submitted to the Ethics Committee. Please correct the information and provide justification for not having a formal approval. Also, please detail how was confidentiality anonymity preserved. Was there any protocol for sensitive issues?
Please discuss ethical issues beyond confidentiality and data protection. Could you better explain the procedures for the protection of the well-being of the research participants given the specificity of the interview guide and sensitive questions such as gender-based violence and voluntary interruption of pregnancy?
Response: The study was conducted in accordance with the Declaration of Helsinki. Ethical review and approval were waived for this study due to the fact that the subjects were invited to participate voluntarily. Informed consent was obtained from all subjects involved in the study. Written informed consent was obtained from the participants to publish this paper.
Results: The results and the discussion sections need to be organized to answer the previously defined objectives. Tables and figures need to be improved.Participants’ characterization could be improved and present more information (region seems to be missing); also consider presenting smaller age groups and social factors comparisons in terms of main outcomes (such as Table 4 – nevertheless age is missing).
Response: Thank you for this observation. A paragraph was included about sociodemographic characteristics has been included.
The discussion is too descriptive. Also, the authors need to better discuss the study's strengths and limitations.
Response: Thank you. Following your recommendation, this section was modified.
Conclusion: Implications of the results for public policy should be enlightened. Specific future recommendations for research and action should be added.
Response: Thank you for this recommendation.
In short, given the data collected, I think there is the potential to do more with the analysis.
Response: Thank you for this suggestion. We aim to publish a new paper with more data about this research.
Round 2
Reviewer 2 Report
Dear authors, thank you for your reply.
I believe you had the opportunity to better improve your manuscript with all the suggestions from the reviewers. Even if not all the suggestions could be considered, at least you could have included some of them in the paper limitations.
There are still many issues that remained unanswered, especially in the methods and results sections. Ethical aspects remain a problem. It is not correct to say: "As the question did not include any personal data, no formal ethical approval was required." when you did collect personal data.
I think your paper still needs revision.
Kind regards.
Author Response
Dear Reviewer:
Authors reviewed all your comments again (Review#1 and Review#2). We hope you find better replies. Thanks in advance:
The authors aim to analyze the access to sexual and reproductive health services during confinement in Spain. The title should mention the study population (women).
Response: We included the word “women”.
This is a promising article. I have some suggestions for your manuscript.
Response: Thank you for this observation.
Introduction: The rationale and hypothesis for the study could be less descriptive and more appealing. A more critical discussion of the links between sexual health (a conceptual definition is missing) and assess to sexual health services and related barriers could be presented in the introduction, which should provide sufficient background and include the most relevant references. Your introduction would benefit from presenting the context of sexual health besides Covid-19 in the specific case of Spain and using proper references citations.
Respuesta: Thank you for this suggestion. This is cross-sectional research aims to know more about these circumstances during the pandemic. Our intention was not to stablish an hypothesis in advance. We focused all this paper on what happened in this specific time in Spain.
A more comprehensive overview of the approaches to sexual and reproductive health diversity of needs across life course and populations would also be expected. It should be inclusive of the needs of women and girls, of different ages, those belonging to national or ethnic groups, and religious and linguistic minorities from different parts of the world with different contexts and experiences. An explicit statement of the objectives being addressed with reference to the concepts, contexts, and population seems to be missing.
Response: We did not find that aspects such as age, ethnicity or religion had an influence on this particular experience. For this reason, the above factors were not discussed.
I would recommend the following articles:
Ferreira-Filho ES, de Melo NR, Sorpreso ICE, Bahamondes L, Simões RDS, Soares-Júnior JM, Baracat EC. Contraception and reproductive planning during the COVID-19 pandemic. Expert Rev Clin Pharmacol. 2020 Jun;13(6):615-622.
Larki M, Sharifi F, Manouchehri E, Latifnejad Roudsari R. Responding to the Essential Sexual and Reproductive Health Needs for Women During the COVID-19 Pandemic: A Literature Review. Malays J Med Sci. 2021 Dec;28(6):8-19.
Mukherjee TI, Khan AG, Dasgupta A, Samari G. Reproductive justice in the time of COVID-19: a systematic review of the indirect impacts of COVID-19 on sexual and reproductive health. Reprod Health. 2021 Dec 20;18(1):252.
Response: We included the first reference because is in line with an idea of the introduction.
The aim of the study needs clarification. In the abstract v. main text. And should be linked to the results.
Response: This was reworded. Please see lines 73-75
Methods need to be described in more detail and clarified. Sampling procedures need more detail. Was there a sampling of both mobile and landline numbers? Also, characterize the survey company and the data collection quality procedures.
Response: Thank you for your observation, this issue was clarified. Please see line 78-80
How was the study presented to the participants; was there an oral informed consent collected and was there a different procedure for participation among younger than 18 years of age participants? Also, provide participation rate and any comparisons between participants and refusals if possible.
Response: Thank you for this observation. All participants were informed about the aims of the research. Informed consent collected was requested and collected prior to participation (lines 269-270).
Please indicate eligibility and exclusion criteria. Was language an exclusion criterion?
Response: We appreciate this observation, that has been included: Please see lines 80-82.
Measure: Please describe the main outcome measures. Which services were considered, how was the access measured, and was the quality of the services evaluated? Was there a comparison question before and during the confinement period to evaluate the effect of covid-19?
Response: This observation is really interesting, but sadly, this could not be assessed and thus we cannot provide information in this sense. This was included in limitations section.
Ethics: Personal data was used (such as the telephone), besides sensitive data (such as nationality, religion, sex life). The project should have been submitted to the Ethics Committee. Please correct the information and provide justification for not having a formal approval. Also, please detail how was confidentiality anonymity preserved. Was there any protocol for sensitive issues?
Please discuss ethical issues beyond confidentiality and data protection. Could you better explain the procedures for the protection of the well-being of the research participants given the specificity of the interview guide and sensitive questions such as gender-based violence and voluntary interruption of pregnancy?
Response: The study was conducted in accordance with the Declaration of Helsinki. Ethical review and approval were waived for this study due to the fact that the subjects were invited to participate voluntarily. Informed consent was obtained from all subjects involved in the study. Written informed consent was obtained from the participants to publish this paper.
Results: The results and the discussion sections need to be organized to answer the previously defined objectives. Tables and figures need to be improved.Participants’ characterization could be improved and present more information (region seems to be missing); also consider presenting smaller age groups and social factors comparisons in terms of main outcomes (such as Table 4 – nevertheless age is missing).
Response: Thank you for this observation. A paragraph was included about sociodemographic characteristics has been included.
The discussion is too descriptive. Also, the authors need to better discuss the study's strengths and limitations.
Response: Thank you. Following your recommendation, this section was modified. Please, see lines 246-253.
Conclusion: Implications of the results for public policy should be enlightened. Specific future recommendations for research and action should be added.
Response: Thank you for this recommendation. Please, see lines 258-263.
In short, given the data collected, I think there is the potential to do more with the analysis.
Response: Thank you for this suggestion. We aim to publish a new paper with more data about this research. Here is the link to the paper published with more information about this study: https://www.mdpi.com/2077-0383/11/13/3777
REVIEW #2
Dear authors, thank you for your reply.
Response: Dear reviewer, we read all your suggestions and comments on revision 1. We considered to include more details in order to reply your requests.
I believe you had the opportunity to better improve your manuscript with all the suggestions from the reviewers. Even if not all the suggestions could be considered, at least you could have included some of them in the paper limitations.
Response: Thank you for this observation. We thought that we included all issues recommended. However, if you consider that we should include more details, just let us know, please.
There are still many issues that remained unanswered, especially in the methods and results sections. Ethical aspects remain a problem. It is not correct to say: "As the question did not include any personal data, no formal ethical approval was required." when you did collect personal data.
As we mentioned in methods section, this survey was conducted by Sigmados. SigmaDos is an international Marketing and Public Survey Study company headquartered in Spain. We asked them to answer this request and the answer was that they do not need the IRB approval because they asked permission participants prior to participate and confidentiality and anonymity was guaranteed.
I think your paper still needs revision.
Response: Authors will be delighted to amend your suggestions, but please, we need more indications about what to change. Thanks in advance.
Kind regards.
Round 3
Reviewer 2 Report
Dear authors, thank you for your reply.
I still believe you can improve your manuscript with all the suggestions from both reviewers.
In my opinion, ethical aspects remain a problem. It is not correct just to say: "As the question did not include any personal data, no formal ethical approval was required."
The aim of the study needs to be better linked with the results and discussion. Why did you focus on gender-based violence if your objective was to analyze the events that took place in relation to access to both contraception methods and to sexual and reproductive health services? If it was not a research goal, ethically sensitive data such as gender-based violence should not have been collected.
I think your paper still needs revision.
Kind regards.
Author Response
Dear reviewer,
Here are our responses to your comments:
In my opinion, ethical aspects remain a problem. It is not correct just to say: "As the question did not include any personal data, no formal ethical approval was required."
Response: Thank you for the suggestion. The sentence was removed.
The aim of the study needs to be better linked with the results and discussion. Why did you focus on gender-based violence if your objective was to analyze the events that took place in relation to access to both contraception methods and to sexual and reproductive health services? If it was not a research goal, ethically sensitive data such as gender-based violence should not have been collected.
Response: We just asked women if they, in their opinion, had experienced some situation of gender-based violence. The response was yes or not. We did not ask more in detail to avoid that other people were hearing her speaking about this issue. Later, this information was related to sociodemographic characteristics.